# Comparative Study of Four Jackfruit Genotypes: Morphology, Physiology and Physicochemical Characterization

David Antonio Morelos-Flores [1], Efigenia Montalvo-González [1], Martina Alejandra Chacón-López [1], Amalio Santacruz-Varela [2], Víctor Manuel Zamora-Gasga [1], Gerardo Torres-García [3,*] and María de Lourdes García-Magaña [1,*]

[1]   Laboratorio Integral de Investigación en Alimentos, Instituto Tecnológico de Tepic/Tecnológico Nacional de México, Tepic C.P. 63175, Nayarit, Mexico
[2]   Programa de Recursos Genéticos y Productividad, Colegio de Postgraduados, Campus Montecillo, Montecillo, Km 36.5 Carretera México-Texcoco, Texcoco C.P. 56230, Estado de México, Mexico
[3]   Centro de Investigación en Alimentos y Desarrollo, Circuito Gobernador C, Ney González#10, Ciudad del Conocimiento, Tepic C.P. 63173, Nayarit, Mexico
*   Correspondence: gerardo.torres@ciad.mx (G.T.-G.); mgarciam@ittepic.edu.mx (M.d.L.G.-M.)

**Abstract:** Jackfruit (*Artocarpus heterophyllus* Lam.) is a climacteric fruit native to India which, due to its edaphoclimatic adaptability, is also found in Mexico, the main exporter of the fruit in Latin America. Despite this, information on the characterization of jackfruit genotypes in Mexico is limited; therefore, the objective of this study was to carry out morphological, physiological, and physicochemical characterization of four jackfruit genotypes, locally known as "Agüitada", "Licenciada", "Rumina", and "Virtud", which are cultivated in Nayarit, Mexico. Morphological analyses revealed 17 traits with significant differences among the genotypes. The respiration rate showed the maximum production of $CO_2$ in the "Agüitada" genotype, with 123.99 mL of $CO_2$ $kg^{-1} \cdot h^{-1}$ at day 2 of storage. The "Rumina" and "Licenciada" genotypes had yellow bulbs while "Agüitada" and "Virtud" had orange bulbs. A principal component analysis revealed different behaviors of the fruits throughout their storage. In general, a wide diversity was revealed among the jackfruit genotypes which are cultivated in the state of Nayarit, Mexico. This study may be useful for their future use in breeding programs.

**Keywords:** *Artocarpus heterophyllus* Lam.; characterization; ethylene production; genotype; respiration rate

## 1. Introduction

Jackfruit is widely cultivated in countries such as India, the Philippines, Pakistan, Sri Lanka, Malaysia, Thailand, and Bangladesh [1]. Due to its edaphoclimatic adaptability, the jackfruit tree can grow on almost any type of soil, but prefers deep, well-drained sandy-loam soils, with plenty of moisture and rich in organic matter. Artocarpus plantations are found on flat or sloping land, on porous soils in tropical areas as well as on light soils. The soils where the crop is to be established must have adequate drainage conditions and must be fertile with water availability, avoiding factors that cause stress, such as excessive heat, wind and frost [2]. Jackfruit is reported to thrive at elevations of 1000 to 1600 m above sea level (a.s.l.) with annual rainfall of 1000 to 2400 mm. Although it is a cold-tolerant species, trees can suffer from severe frost damage, so a warm-humid climate with temperatures between 19 and 29 °C is best [3]. The 50 known species of the genus Artocarpus are mostly humid tropical trees, originating from areas with moderate monsoon climates and a short dry season [2]. The propagation of the fruits is by seed, cuttings or grafting, and these methods can be carried out at any time of the year. These fruits are also found in some countries in the Americas, such as Brazil, United States, and Mexico [2]. In Mexico, the state of Nayarit is the main producer of jackfruit, contributing more than 90% of the national production [4]. Thus, jackfruit is one of the most important crops in the state, competing

with traditional crops such as mango and banana; however, limited information is available on the characteristics of the fruit.

Studies on jackfruit from other countries have reported that the leaves are dark green and can be elliptical, oblong, or ovate. Moreover, the leaves have a petiole of dark green color with dimensions of 1 to 5 cm [5]. The fruit is cataloged as having rough peel and conical-hexagonal carpel apices, which turn from green to dark yellow during ripening [6]. Jackfruit is a large fruit and has a strong peduncle because it can weigh more than 50 kg. It has a fibrous central rachis [1], and the perianths are the largest constituents, along with regions that bear seeds and the aril [7]. The fleshy aril, known as the bulb, comprises between 20% and 30% of the edible portion of the fruit; it is golden yellow (depending on the genotype) and has a peculiar flavor, [8]. Inside the bulbs, the seeds are light brown in color with a spheroid shape and are enclosed in a thin white membrane [3] (see Figure 1).

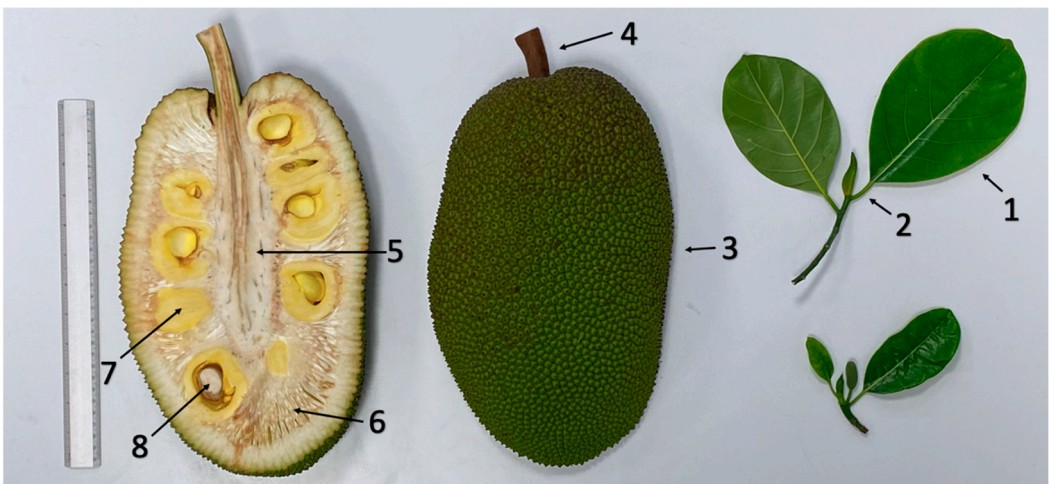

**Figure 1.** Morphological aspect of jackfruit at physiological maturity (1: leaves, 2: petiole, 3: peel, 4: peduncle, 5: rachis, 6: perianths, 7: bulb and 8: seed).

Ranasinghe et al. [9]. reported that the pulp, leaves, and peel of jackfruit have been used in traditional medicine as, for example, anti-inflammatory, anti-proliferative, hypoglycemic, antioxidant, and antimicrobial agents; however, those authors only described the characteristics for certain types of jackfruit. Jackfruit has a high content of phytochemicals, mainly phytosterols, carotenoids, and phenolic compounds [10], which have been found to exhibit a high antioxidant capacity [11]; however, the genotypes analyzed in that study were not specified.

Most studies on this fruit are from Asia (mainly India) and have reported important morphological, physiological, and chemical variations among genotypes of jackfruit, such as the shape, size, and color of the fruit, the size of the leaves, sensory quality, and pulp color [12–14]. Other important parameters considered in studies of jackfruit include the respiration rate (RR) and ethylene production (EP) [7,15,16], because during the ripening process, various changes manifest that are interesting with regard to the sensory quality of the fruit (color, total soluble solids, texture, acidity, volatile compounds, among others) [17]. However, information on the characterization of jackfruit genotypes in Mexico and the rest of Latin America is limited [12–14]. Nayarit is the main producer of jackfruit in Mexico and different genotypes of the fruit grow there, providing an opportunity to study its characteristics; therefore, the objective of this research was to study the morphological, physiological, and physicochemical characteristics of jackfruit genotypes in the region to allow a better use of the fruit.

## 2. Materials and Methods

### 2.1. Biological Material

In this study, four genotypes were used, known locally as "Agüitada", "Rumina", "Licenciada", and "Virtud", harvested at physiological maturity in February 2022 at Estacion Nanchi (21°47′25.3″ N 105°03′41.6″ W), municipality of Santiago Ixcuintla, Nayarit, Mexico, at 32 m a.s.l. Fifty leaves and 45 fruits were used for the morphological, physiological, and physicochemical analyses, i.e., 15 of each jackfruit (*Artocarpus heterophyllus* Lam.) genotype. Subsequently, the fruits were transported to the Laboratorio Integral de Investigacion en Alimentos of the Tecnologico Nacional de Mexico-Tepic, where they were washed with tap water and an antifungal treatment with thiabendazole at 800 ppm was applied by immersion for 3 min. The fruits were then left to air-dry at room temperature. Subsequently, the peduncle was sealed with copper oxychloride to prevent the entry of pathogens. Finally, the fruits were stored in a refrigeration chamber at 25 °C and a relative humidity of 90%.

### 2.2. Morphological Analysis

A morphological characterization of the leaves, fruits, bulbs, and seeds was performed according to guidelines proposed by the International Plant Genetic Resources Institute [18] for jackfruit. The morphological characteristics considered in this study were as follows: length of the petiole (LP), length of the peduncle (LPE), equatorial diameter of the peduncle (DEPE), weight of the peel (WP), thickness of the peel (TP), length of the rachis (LR), rachis width (RW), bulb weight (BW), bulb length (BL), bulb width (BWI), bulb thickness (BT), seed length (SL), seed width (SW), number of seeds per fruit (NSPF), weight of seedless bulbs (WSB), leaf length (LL), leaf width (LW), equatorial diameter of the fruit (EDF), and polar diameter of the fruit (PDF). A Vernier caliper was used to obtain these measurements, and a tape measure was used to measure greater lengths.

### 2.3. Physiological Analysis

To evaluate the respiration rate (RR) and ethylene production (EP) of the jackfruit, the method proposed by Tovar et al. [19] was followed, with some modifications. Whole fruits (genotypes "Agüitada", "Licenciada" and "Virtud") were placed in 20 L containers. The genotype "Rumina" was placed in a 40 L container due to its larger size; physiological analyses were performed every 24 h, and on each occasion for RR and EP analysis, the containers in which the fruits were placed were hermetically sealed for 30 min before 0.5 mL was extracted from the headspace. Samples were analyzed using a gas chromatograph (GC6890; Hewlett-Packard, Palo Alto, CA, USA) equipped with an HP-PlotQ column (15 m × 0.53 mm and 40 μm film thickness), a thermal conductivity detector to detect $CO_2$ and a flame ionization detector to detect ethylene. The temperature of the injection port and of both detectors was 250 °C; the oven temperature ramps were from 50 °C to 80 °C with a rate of change of 30 °C $min^{-1}$. $H_2$ was used as carrier gas at a flow rate of 30 mL $min^{-1}$. The results obtained for RR are reported as mL of $CO_2$ $kg^{-1} \cdot h^{-1}$ and as μL $kg^{-1} \cdot h^{-1}$ for EP. RR and EP fruits were also used to determine physiological weight loss (PWL) using a digital scale (L-PCR; Torrey, Mexico City, Mexico); the difference in weight with respect to the original value was expressed as a percentage [15]; the following equation was applied:

$$\%PWL = \frac{original\ weight - current\ weight}{original\ weight} \times 100$$

Once the analysis was complete, the fruits were stored at room temperature and physiological analyses were continued every 24 h until fruit senescence.

### 2.4. Physicochemical Analysis

An analysis of total soluble solids (TSS; method 932.14) was performed using a pocket digital refractometer (3810 PAL-1; ATAGO, Tokyo, Japan). Titratable acidity (TA; method

942.15) was determined using an automatic titrator (TritoLine Easy; Schott Instruments, Bad Gandersheim, Germany), and pH was measured (method 981.12) using a potentiometer (HI 2210; Hanna Instruments, Woonsocket, RI, USA), in jackfruit bulbs [20]. Firmness was evaluated following the method proposed by Morelos-Flores et al. [16], using a texture meter (TA.TXplus; Stable Micro Systems, Godalming, UK) with a 3-mm tip for the peel and a 2-mm tip for the bulbs at three points along the fruit (upper, middle, and bottom); values are reported in Newtons (N). A colorimeter (NH300; M&A Instruments, Arcadia, CA, USA) was used to determine the color of the peel and bulbs, reporting the angle (° hue), chroma, and luminosity. Physicochemical analyses were carried out on days 1, 3, 5, 7, 9, and 11 of storage.

### 2.5. Statistical Analysis

Morphological, physiological, and physicochemical characterization data were subjected to a two-way ANOVA (genotype and days of storage) following a Tukey test for comparisons of means ($p \leq 0.05$); these analyses were performed with Prism 9 software (GraphPad, San Diego, CA, USA). Principal component analysis (PCA) was performed considering as multiple factors both the four evaluated genotypes and three days of storage (initial, middle, and final day). In addition, the response variables described in this analysis were those obtained from the physicochemical and physiological analyses. PCA was carried out with Statistica software (version 12; StatSoft, Tulsa, OK, USA).

## 3. Results

### 3.1. Morphological Analysis

The results for the morphological analyses (Figure 2, Table 1) revealed 17 traits with significant differences ($p \leq 0.05$). In contrast, BL and BT were statistically similar in all genotypes. LP was similar for the "Rumina" and "Virtud" genotypes, with these varieties showing the highest values (2.3 and 2.16 cm, respectively). "Agüitada" and "Licenciada" differed from the other genotypes.

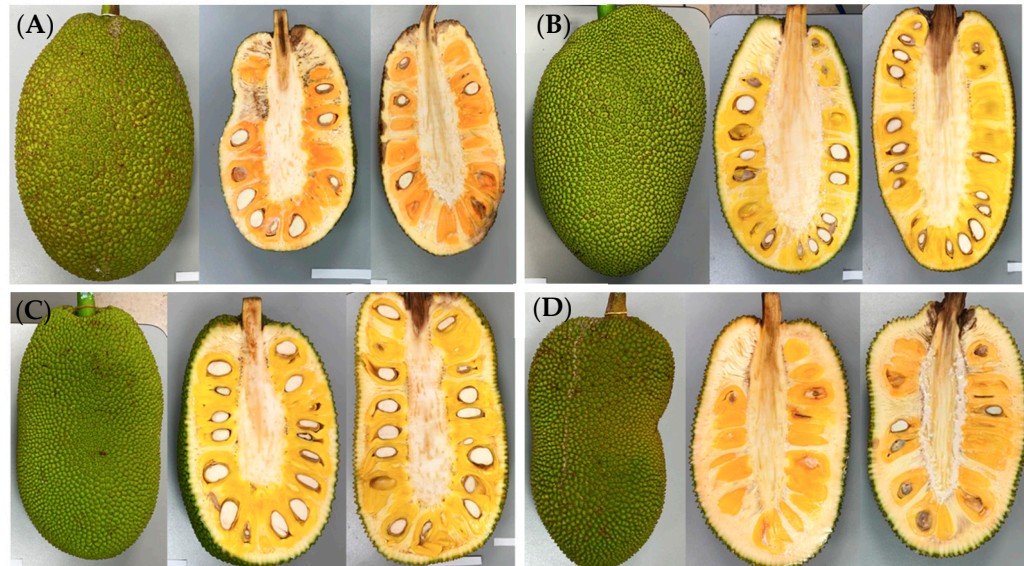

**Figure 2.** Cross-section of the different jackfruit genotypes showing morphological aspect after five days of storage. (**A**): Agüitada genotype, (**B**): Rumina genotype, (**C**): Licenciada genotype, (**D**): Virtud genotype; Storage condition: 25 °C, 90% relative humidity).

With regard to LL, the "Agüitada" genotype was the only one that presented significant differences in relation to the other genotypes, with a shorter leaf (12.05 cm). Moreover, in LW, "Agüitada" and "Virtud" presented significant differences ($p \leq 0.05$), with 5.96 cm being the narrowest and 8.63 cm the widest, respectively. Several studies have reported

LL and LW as response variables for the characterization of jackfruit genotypes [13,14]. Regarding LPE, "Rumina" and "Virtud" gave the highest value (33.24 cm), while the lowest value was found with "Licenciada" (19.22 cm). "Virtud" presented statistical differences with regard to DEPE (8.38 cm) compared with the other genotypes, while "Agüitada" and "Rumina" were similar. On other hand, "Rumina" had the highest WP (2.83 kg), followed by "Licenciada" and, finally, "Agüitada" (1.17 kg). For TP, "Virtud" was the only genotype with statistical differences from the others, with the greatest thickness (2.12 cm). For LR and RW, "Rumina" had the largest dimensions (27.38 and 8.74 cm, respectively). "Agüitada" was similar to "Licenciada" and "Virtud" (in RW), but "Licenciada" and "Virtud" were different.

**Table 1.** Morphological Analysis of Four Jackfruit Genotypes from the State of Nayarit, Mexico.

| Morphological Parameter | "Agüitada" | "Rumina" | "Licenciada" | "Virtud" |
|---|---|---|---|---|
| Length of the petiole (cm) | 1.93[a] | 2.31[b] | 1.74[c] | 2.16[b] |
| Length of the peduncle (cm) | 25.54[a] | 33.24[b] | 19.22[c] | 30.52[ab] |
| Equatorial diameter of the peduncle (cm) | 6.60[a] | 9.85[a] | 7.26[b] | 8.38[c] |
| Weight of the peel (kg) | 1.72[a] | 2.83[b] | 2.51[bc] | 2.12[ac] |
| Thickness of the peel (cm) | 1.21[a] | 1.27[a] | 1.27[a] | 2.08[b] |
| Length of the rachis (cm) | 20.60[a] | 27.38[b] | 24.35[c] | 19.91[a] |
| Rachis width (cm) | 6.50[a] | 8.74[b] | 6.65[a] | 4.11[c] |
| Bulb weight (kg) | 2.72[a] | 5.40[b] | 4.58[b] | 1.77[a] |
| Bulb length (cm) | 3.49[a] | 3.25[a] | 3.26[a] | 3.16[a] |
| Bulb width (cm) | 5.23[a] | 6.03[b] | 5.99[b] | 5.06[a] |
| Bulb thickness (cm) | 0.72[a] | 0.90[a] | 0.89[a] | 0.85[a] |
| Seed length (cm) | 3.02[a] | 3.10[a] | 3.60[b] | 3.34[c] |
| Seed width (cm) | 2.30[a] | 2.36[ab] | 2.51[b] | 2.30[a] |
| Number of seeds per fruit | 89.70[a] | 130.33[b] | 101.17[a] | 52.10[c] |
| Weight of seedless bulbs (kg) | 2.41[a] | 4.63[b] | 3.75[c] | 1.57[a] |
| Leaf length (cm) | 12.05[a] | 13.90[b] | 14.08[b] | 14.13[b] |
| Leaf width (cm) | 5.96[a] | 7.78[b] | 8.50[c] | 8.63[c] |
| Equatorial diameter of the fruit (cm) | 62.07[a] | 69.30[b] | 69.97[b] | 66.83[b] |
| Polar diameter of the fruit (cm) | 83.27[a] | 98.17[bc] | 101.27[b] | 93.63[c] |

The values are presented as the means ± standard deviation. Lowercase letters represent the effect between jackfruit genotypes. Different letters indicate significant differences ($p \leq 0.05$) between genotypes.

The EDF and PDF results show that "Agüitada" fruits were the smallest (62.07 and 83.27 cm, respectively) and the only ones to show a statistical difference ($p \leq 0.05$) from the other genotypes. The "Rumina" and "Licenciada" genotypes predominated in size, with EDF of 69.30 and 69.97 cm and a PDF of 98.17 and 101.27 cm, respectively. In the bulbs, BL and BT were similar among all genotypes. "Rumina" and "Licenciada" showed the highest values for BW (5.4 and 4.58 kg, respectively), while "Agüitada" and "Virtud" were similar for both genotypes. Concerning NSPF, no significant differences were found; however, "Rumina" showed the highest value (130.33 seeds). "Rumina" presented the highest WSB (4.63 kg), which was significantly different from the other. WSB was similar in "Agüitada" and "Virtud".

*3.2. Physiological Analysis*

Firstly, we observed differences during the final days of storage of the fruits. The "Agüitada" and "Virtud" genotypes were able to maintain their optimum consumption qualities 9 days after harvest, while "Rumina" and "Virtud" were reached 11 days.

The day on which the climacteric peak appeared differed among the genotypes, starting with "Agüitada", which had the highest RR of 123.99 mL of mL of $CO_2$ kg$^{-1} \cdot$h$^{-1}$ on day 2 of storage; the fruits from "Virtud" had their climacteric peak on day 4 of storage with values of 69.91 mL of $CO_2$ kg$^{-1} \cdot$h$^{-1}$, and the "Rumina" and "Licenciada" genotypes presented their climacteric peak on the same day of storage (day 6), with an average RR of 93.8 and 71.23 mL of $CO_2$ kg$^{-1} \cdot$h$^{-1}$, respectively; ($p \leq 0.05$). On the other hand, in the

evaluation of storage days, significant differences were found ($p \leq 0.05$) in all genotypes. Fruits belonging to the genotype "Agüitada" showed an increase in RR after the onset of the climacteric peak, while the remaining genotypes only showed fluctuations. The RR was 53.16, 41.53, and 31.74 mL of $CO_2$ $kg^{-1} \cdot h^{-1}$ for "Agüitada", "Virtud", and "Licenciada", respectively. The "Rumina" and "Licenciada" genotypes lasted for up to 11 days of storage, preserving optimal physical characteristics for marketing, in contrast to "Agüitada" and "Virtud", which remained in good condition until 9 days after harvesting.

Data obtained in the EP analysis (Table 2) showed significant differences ($p \leq 0.05$) among the genotypes. On the initial day of storage, ethylene was not detected in the harvested fruits, as reported previously [16]; however, on day 2 of storage, the "Virtud" genotype presented statistically significant differences ($p \leq 0.05$) compared to the others, showing a maximum EP of 84.36 $\mu L\ kg^{-1} \cdot h^{-1}$. The "Rumina" and "Licenciada" genotypes showed statistically significant differences ($p \leq 0.05$), with a maximum EP of 51.87 and 40.43 $\mu L\ kg^{-1} \cdot h^{-1}$, respectively, on day 4. Similarly, the fruits belonging to the "Agüitada" genotype presented their maximum EP on day 4 with an average value of 63.92 $\mu L\ kg^{-1} \cdot h^{-1}$. In the analysis of storage days, significant differences ($p \leq 0.05$) were found in all jackfruit genotypes. These results demonstrate a close relationship with RR, since the genotypes also showed an increase in EP after the climacteric peak appeared, showing different values up to fruit senescence.

In the evaluation of PWL (Table 2), no statistically significant differences were found ($p \geq 0.05$) at the end of the shelf life of the fruits. In contrast, in the analysis of storage days, significant differences ($p \leq 0.05$) were found in all of the evaluated genotypes. The fruits belonging to the genotype "Licenciada" showed the highest PWL with 9.39% of the total weight lost, while "Virtud" genotype showed a loss of 8.63%.

Although the limits for the PWL of jackfruit have not yet been established for commercialization, some authors have reported losses >11% PWL in control fruits stored for 10 days at 20 °C and for 8 days at 25 °C [15,16]. In this study, losses were close to 10%, representing an acceptable percentage.

**Table 2.** Respiration rate, ethylene production rate, and physiological weight loss.

| Storage Days (25 °C) | Agüitada | Rumina | Licenciada | Virtud |
|---|---|---|---|---|
| Respiration rate (mL of $CO_2$ $kg^{-1} \cdot h^{-1}$) | | | | |
| 1 | 53.17[aA] | 71.40[bA] | 32.08[cAD] | 42.11[cA] |
| 2 | 125.53[aB] | 60.31[bA] | 52.35[bBD] | 56.48[bAB] |
| 3 | 118.37[aB] | 66.76[bA] | 66.92[bC] | 66.38[bBC] |
| 4 | 112.78[aB] | 77.81[bcA] | 63.59[bBC] | 80.02[cC] |
| 5 | 74.30[aC] | 69.71[aA] | 51.37[bBCD] | 63.81[abBC] |
| 6 | 75.80[aC] | 96.62[bB] | 69.20[acBC] | 60.55[cB] |
| 7 | 58.97[aAC] | 70.31[aA] | 24.63[bA] | 58.16[aAB] |
| 8 | 61.01[aAC] | 64.47[aA] | 42.37[bD] | 53.99[abAB] |
| 9 | 69.47[aAC] | 65.49[abA] | 44.82[cD] | 52.90[bcAB] |
| 10 | - | 65.75[aA] | 47.52[bD] | - |
| 11 | - | 93.43[aB] | 59.54[bBC] | - |
| Ethylene production ($\mu L\ kg^{-1} \cdot h^{-1}$) | | | | |
| 1 | 0.00[aA] | 0.00[aA] | 0.00[aA] | 0.00[aAD] |
| 2 | 10.16[aBF] | 5.04[aAB] | 7.43[aBC] | 23.52[bA] |
| 3 | 47.25[aC] | 6.81[bBD] | 6.06[bBC] | 84.56[cB] |
| 4 | 63.31[aD] | 54.92[bC] | 40.45[cF] | 27.15[dA] |
| 5 | 49.52[aC] | 30.87[bE] | 17.75[cD] | 19.10[cCF] |
| 6 | 32.90[aE] | 18.80[bF] | 19.7[bD] | 11.10[cDFE] |
| 7 | 27.92[aE] | 26.03[aE] | 9.30[bBE] | 14.74[cF] |
| 8 | 14.23[aF] | 17.17[aF] | 16.24[aD] | 5.68[bEG] |
| 9 | 3.83[aAB] | 8.62[aBG] | 4.13[aAC] | 3.83[aG] |
| 10 | - | 7.72[aBG] | 2.23[bA] | - |
| 11 | - | 5.36[aADG] | 2.37[aA] | - |

**Table 2.** *Cont.*

| Storage Days (25 °C) | Agüitada | Rumina | Licenciada | Virtud |
|---|---|---|---|---|
| | Physiological weigth loss (%) | | | |
| 1 | 0.00$^{aA}$ | 0.00$^{aA}$ | 0.00$^{aA}$ | 0.00$^{aA}$ |
| 2 | 1.12$^{aB}$ | 0.88$^{aB}$ | 1.08$^{aB}$ | 1.00$^{aB}$ |
| 3 | 2.62$^{aC}$ | 1.82$^{bC}$ | 2.05$^{abC}$ | 2.13$^{abC}$ |
| 4 | 3.97$^{aD}$ | 2.83$^{bD}$ | 3.25$^{abD}$ | 3.39$^{abD}$ |
| 5 | 5.78$^{aE}$ | 4.38$^{bE}$ | 4.86$^{bE}$ | 4.49$^{bE}$ |
| 6 | 5.90$^{aE}$ | 4.90$^{bE}$ | 5.22$^{abE}$ | 5.60$^{abF}$ |
| 7 | 6.90$^{aF}$ | 5.88$^{bF}$ | 6.16$^{abF}$ | 6.69$^{aG}$ |
| 8 | 8.06$^{aG}$ | 6.90$^{bG}$ | 7.12$^{cG}$ | 7.85$^{acH}$ |
| 9 | 8.92$^{aG}$ | 7.53$^{bG}$ | 7.72$^{bG}$ | 8.63$^{aH}$ |
| 10 | - | 8.57$^{aH}$ | 8.85$^{aH}$ | - |
| 11 | - | 9.39$^{aH}$ | 9.70$^{aH}$ | - |

Values are presented as means ± standard deviation (*n* = 3). Lowercase letters represent the effect between jackfruit genotypes and capital letters indicate effect for days. Different letters indicate significant differences ($p \leq 0.05$) between genotypes or days.

### 3.3. Physicochemical Analysis

As mentioned above, differences among genotypes were observed in the final days of storage. "Agüitada" and "Virtud" were able to maintain their eating qualities 9 days after harvesting, while "Rumina" and "Virtud" reached 11 days, as can be seen in the results of the physicochemical analyses (Table 3). Physicochemical analyses, including peel color (PC), bulb color (BC), peel firmness (PF), bulb firmness (BF), and TA, showed significant differences ($p \leq 0.05$). For PC, no statistically significant differences were found ($p \geq 0.05$) at the beginning or end of the shelf life of the fruits. In the opposite case, significant differences ($p \leq 0.05$) were found in the storage days of the genotypes. On the first day of storage, the maximum value for PC was found in "Licenciada" (92.69 °Hue) and the minimum in "Rumina", with 86.82 °Hue; these values corresponded to a pear green color. At the end of their shelf life, the final maximum value was 78.86 °Hue for "Licenciada" and the minimum value was 69.22 °Hue for "Agüitada", which showed olive green colorations.

In the analysis of BC, significant differences were observed between the genotypes and their storage day, with "Agüitada" and "Virtud" in one group and "Rumina" and "Licenciada" in another, based on the color of the pulp (orange and yellow). On the first day of storage, the BC values of "Agüitada" and "Virtud" were 67.55 and 72.84 °Hue, respectively, giving a mustard yellow color. On the other hand, the BC values for "Rumina" and "Licenciada" were 80.44 and 81.89 °Hue, respectively, giving them a pineapple yellow color. At the end of their shelf life, the BC values for "Agüitada" and "Virtud" were 63.28 and 66.86 °Hue, respectively ($p \leq 0.05$), giving them a saffron yellow color. "Rumina" and "Licenciada" had statistically similar values ($p \geq 0.05$) of 76.04 and 77.31 °Hue, respectively, giving them a butter yellow color. The differences between "Agüitada" and "Virtud" versus "Rumina" and "Licenciada" were statistically different ($p \leq 0.05$).

Significant differences in PF ($p \leq 0.05$) were observed only in the "Rumina" genotype, but there were significant differences in the storage day of all genotypes. On day 1 of storage, the PF values were 348.79 N ("Licenciada"), 308.38 N ("Agüitada"), and 409.15 N ("Rumina"); however, at the end of shelf life, "Agüitada" and "Licenciada" (215.58 and 220.55 N) showed statistical differences ($p \leq 0.05$) with respect to "Rumina" and "Virtud" (293.45 and 289.16 N). On the other hand, in BF, significant statistical differences ($p \leq 0.05$) were found in all genotypes. On day 1 of storage, "Rumina" showed the maximum value (42.51 N) while "Virtud" had the lowest (15.86 N). At the end of shelf life, no significant differences were found among genotypes, and values ranged from 4.72 N ("Rumina") to 2.96 N ("Licenciada").

**Table 3.** Physicochemical Analysis of Peel and Bulbs of Jackfruit Genotypes.

| Storage Days (25 °C) | "Agüitada" | "Rumina" | "Licenciada" | "Virtud" |
|---|---|---|---|---|
| Peel color (°H) | | | | |
| 1 | 90.58[aA] | 86.82[aA] | 092.69[aA] | 88.38[aA] |
| 3 | 86.93[aAB] | 88.86[aA] | 92.94[aA] | 86.45[aA] |
| 5 | 79.55[aB] | 84.25[abAB] | 91.42[bAB] | 82.34[aA] |
| 7 | 78.20[aB] | 82.72[aAB] | 83.49[aBC] | 81.82[aA] |
| 9 | 69.22[aC] | 71.23[aC] | 81.88[bC] | 72.25[aB] |
| 11 | – | 72.30[abBC] | 78.86[aC] | – |
| Bulbs color (°H) | | | | |
| 1 | 67.55[aA] | 80.44 ± 2.23[bA] | 81.89[bA] | 72.84[aA] |
| 3 | 66.87[aA] | 79.61[bA] | 79.64[bAB] | 70.12[aAB] |
| 5 | 62.32[aB] | 77.70[bA] | 77.00[bB] | 69.57[cAB] |
| 7 | 60.29[aB] | 77.36[bA] | 75.16[bB] | 68.86[cAB] |
| 9 | 63.28[aAB] | 76.91[bA] | 76.17[bB] | 66.86[aB] |
| 11 | – | 76.04[bA] | 77.31[bAB] | – |
| Peel firmness (N) | | | | |
| 1 | 308.38[aA] | 409.15[bA] | 348.79[abA] | 343.92[aA] |
| 3 | 305.30[aA] | 421.12[bAB] | 307.81[aAC] | 321.89[aAB] |
| 5 | 295.75[aAB] | 408.90[bAB] | 298.00[aAC] | 315.23[aAB] |
| 7 | 232.19[acBC] | 349.18[bB] | 230.18[cB] | 293.30[abAB] |
| 9 | 215.58[aC] | 250.03[abC] | 221.98[aCB] | 289.16[bB] |
| 11 | – | 293.45[bC] | 220.55[aB] | – |
| Bulb firmness (N) | | | | |
| 1 | 34.27[aA] | 42.51[bA] | 26.52[cA] | 15.86[dA] |
| 3 | 19.10[aB] | 17.90[aB] | 21.58[aB] | 10.69[aB] |
| 5 | 3.85[aC] | 12.67[aB] | 7.76[aC] | 5.62 [aC] |
| 7 | 3.56[aC] | 6.91[aC] | 6.35[aCD] | 4.46[aC] |
| 9 | 3.48[aC] | 5.41[aC] | 4.67[aCD] | 3.87[aC] |
| 11 | – | 4.72[aC] | 2.96[aD] | – |
| Titratable acidity (%) | | | | |
| 1 | 0.10[aA] | 0.06[aA] | 0.11[aA] | 0.29[bA] |
| 3 | 0.43[acB] | 0.48[aB] | 0.25[bB] | 0.41[cB] |
| 5 | 0.33[aC] | 0.41[bB] | 0.26[aB] | 0.29[aA] |
| 7 | 0.25[aD] | 0.30[aC] | 0.17[bA] | 0.28[aA] |
| 9 | 0.25[aD] | 0.27[aC] | 0.21[aB] | 0.22[aA] |
| 11 | – | 0.23[aC] | 0.21[aB] | – |
| pH | | | | |
| 1 | 5.82[aA] | 6.25[aA] | 5.52[aA] | 4.49[bA] |
| 3 | 4.35[aB] | 4.35[aB] | 4.18[aA] | 4.39[aA] |
| 5 | 4.55[aB] | 4.17[aB] | 4.67[aA] | 4.49[aA] |
| 7 | 5.41[aAC] | 5.20[aC] | 5.59[aB] | 5.31[aB] |
| 9 | 5.30[aC] | 5.27[aC] | 5.51[aB] | 5.29[aB] |
| 11 | – | 5.11[aC] | 4.90[aA] | – |
| Total soluble solids (TSS) | | | | |
| 1 | 4.85[aA] | 3.70[aA] | 8.6[bA] | 13.13[cA] |
| 3 | 20.28[aB] | 19.38[aB] | 16.85[bB] | 20.60[aB] |
| 5 | 25.38[aC] | 23.55[aC] | 22.30[aC] | 21.06[bB] |
| 7 | 26.45[aC] | 26.15[aC] | 21.57[bC] | 23.73[aBC] |
| 9 | 27.46[aC] | 26.20[aC] | 23.01[bC] | 25.57[aC] |
| 11 | – | 26.83[aC] | 23.05[bC] | – |

Values are presented as means ± standard deviation (*n* = 3). Lowercase letters represent the effect between jackfruit genotypes and capital letters indicate effect for days. Different letters indicate significant differences (*p* ≤ 0.05) between genotypes or days.

There were statistical differences (*p* ≤ 0.05) in TA in the genotypes on day 1 of storage, with values ranging from 0.06% for "Rumina" to 0.29% for "Virtud". During storage, all

genotypes began to generate TA values without significant differences ($p \geq 0.05$) until the final day, when values ranged from 0.21% ("Licenciada") to 0.25% ("Agüitada"). The genotypes that were evaluated showed significant differences ($p \leq 0.05$) with respect to AT and storage days; however, the "Virtud" genotype only showed differences on day 3 of storage (0.41%). The pH did not show significant differences in the jackfruit genotypes evaluated in this study; nevertheless, significant differences were found in the evaluation of storage days. At the beginning of the test, pH values of the bulbs ranged from 4.49 ("Virtud") to 6.25 ("Rumina"); at the end of their shelf life, they remained at 4.90 ("Licenciada") to 5.30 ("Agüitada").

With regard to TSS, one of the most important parameters of the fruits, significant differences were found in the genotypes evaluated and in their storage days ($p \leq 0.05$). On day 1 of storage, "Virtud" (13.13 TSS) and "Licenciada" (8.6 TSS) showed the highest values, with "Rumina" (3.70 TSS) showing the lowest. At the end of storage, "Agüitada" (27.46 TSS) was the genotype with the highest values of this parameter, followed by "Rumina" (26.83 TSS) and "Virtud" (25.57 TSS). Finally, "Licenciada" (23.05 TSS) had the lowest TSS production.

### 3.4. Principal Component Analysis

The selection of variables used for PCA (Figure 3A) explained 75.21% of the variability. This projection indicates that PCA 1 (53.30%) had a correlation with BF, PWL, PC, pH, BC, and PF. On the other hand, TSS, TA, EP, and RR had correlations in the negative zone. During the fruit ripening process, BF, PWL, PC, pH, BC, and PF are considered the most important parameters in the commercial industry, as they will determine fruit quality [21]. First, PF and BF are closely related, as they are regulated by the enzymes polygalacturonase, cellulase, and β -galactosidase [22]. This is related to the loss of cell firmness and turgor due to the loss of intracellular water, generating PWL. Finally, due to the loss of firmness and the decompartmentalization of the cells, a mixing of organic acids and other cell components occurs, generating changes in pH. Finally, the PC parameter is regulated by chlorophyllase activity and, in the case of jackfruit bulbs, the generation of its characteristic coloration is due to enzymatic activity producing β and α carotene [16]. These correlations among variables show an expected pattern for climacteric fruit. As mentioned above, RR, EP, TA, and TSS are correlated, since respiration uses starches, sugars, and organic acids as substrates, generating reserve losses [23]. At the same time, the activity of ACC oxidase is oxygen-dependent, having a close relationship with RR [17]. It is also known that ethylene production coordinates the generation of enzymes that will initiate fruit ripening; thus, the relationship between AT, TSS, and EP is evident.

In the case of PCA 2 (21.91%), the values of EP, TA, PF, PC, PWL, and BC indicated correlations. In contrast, TSS, BF, and RR did not show significant changes in PCA 2. Finally, only the pH variable was in the negative zone. PCA 2 showed a correlation between EP with the physicochemical parameters mentioned above, which corresponds to an expected trend, as the processes involved in fruit ripening are regulated by EP.

Regarding the projection of cases (Figure 3B), it is possible to observe a clustering of the genotypes corresponding to the same day of analysis. Fruits of storage day 1 were positioned in the first and fourth quadrants, which indicates a greater affinity for PCA 1 to explain their response. On the other hand, fruits at day 5 of storage were positioned in the second quadrant of the plane, indicating an inverse relationship in the explanation of the response attributed to PCA 1. For the final day of storage, the genotypes were in the third quadrant of the plane, similarly showing a greater relationship with PCA 1 and 2.

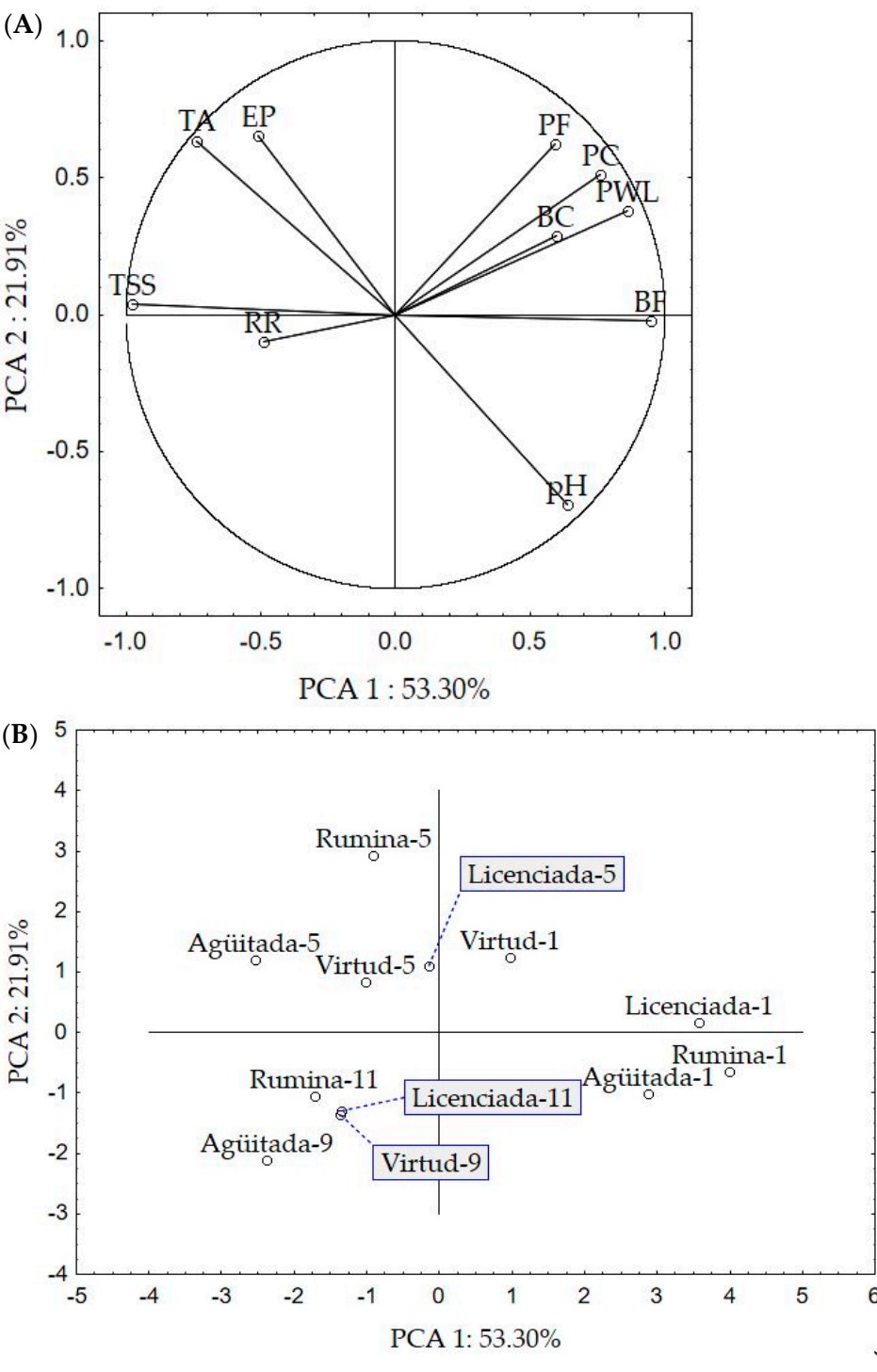

**Figure 3.** Principal component analysis of jackfruit genotypes evaluated on different days of storage; projection of variables (**A**) and projection of cases (**B**).

## 4. Discussion

### 4.1. Morphological Analysis

The results from our morphological analysis can be compared with those of Chhetri et al. [14], who evaluated 40 jackfruit genotypes from northeast India (Arunachal Pradesh) and reported a mean LP of 1.96 cm, and 16.74 cm and 5.73 cm for LR and RW, respectively. Moreover, in WP, different values were reported by Ibrahim et al. [24]; they measured 10 jackfruit genotypes from four different areas of Bangladesh (Rajshahi, Natore, Noagoan, and Chapai Nawabgonj) and reported a general average of 0.77 kg. In the case of TP, Kavya

et al. [25], using 20 jackfruit genotypes from Karnataka, India, reported an average value of 1.66 cm. On the other hand, Akter and Rahman [26] evaluated the WP in 23 jackfruit genotypes from Jamalpur, a locality in Bihar, India. The highest value was 6.81 kg per fruit, which is higher than those found in this study in "Rumina". Karunakaran et al. [27] reported results from three jackfruit genotypes from Tumkur, Karnataka, India, with a mean BW of 0.737 kg (CHESHJF 1), 0.754 kg (CHESHJF 2), and 4.5 kg (CHESHJF 3) for the genotypes evaluated over 3 years.

Data obtained from jackfruit leaves can be compared with those obtained by Biswajit and Kartik [13]. They evaluated six jackfruit genotypes from six locations in Assam, India, and reported average values 15.19 and 8.58 cm for LL and LW, respectively. Singh et al. [28] investigated 42 jackfruit genotypes from different agroclimatic zones in Tripura, India, and reported values ranging from 8.5 to 17.8 cm (LL) and from 4 to 10 cm (LW) in leaves. On the other hand, Chhetri et al. [14] reported values of 14.30 cm (LL) and 7.93 cm (LW), and Chandrashekar et al. [29] reported values of 16.73 cm (LL) and 7.87 cm (LW).

Akter and Rahman [26] reported average values for the dimensions of jackfruit genotypes of 37.54 cm (EDF) and 62.8 cm (PDF). On the other hand, Singh et al. [28] reported average values of 45.16 cm (EDF) and 76.30 cm (PDF) cm, and Biswajit and Kartik [13] reported values of 28.67 cm (EDF) and 57 cm (PDF) cm. Chhetri et al. [14] reported means of 16.34 cm (EDF) and 47.66 cm (PDF).

There is therefore statistical evidence that variability among jackfruit genotypes is present in the parameters evaluated in this research (Anu et al. [30], Mitra and Maity [31], Rahman et al. [32], Rai et al. [33], Reddy et al. [34], and Singh et al. [35]). This variability in the morphological attributes of fruits could be due to cross-pollination and different agroclimatic conditions [12].

*4.2. Physiological Analysis*

Respiration is one of the most important physiological processes of fruits, as it determines the storage life and quality. Respiration produces compounds necessary to determine the speed of metabolic processes related to parameters of interest, such as aroma, firmness, and flavor [36]. Similarly, energy consumption is necessary for the degradation of chlorophyll, conversion of starch into sugar, and degradation of the cell wall, among other processes [37]. Data on jackfruit RR are available from different parts of the world. Saxena et al. [7] reported a climacteric peak of 166 mL of $CO_2$ $kg^{-1} \cdot h^{-1}$ for jackfruits (genotype not specified) from Mysore, Karnataka, India. Mata-Montes de Oca et al. [15] reported RR values of 90.7 mL of $CO_2$ $kg^{-1} \cdot h^{-1}$ at climacteric peak in jackfruit (genotypes not specified) harvested in Ixtapa de la Concepción, Compostela, Nayarit, Mexico; however, Morelos-Flores et al. [16] reported for RR values at climacteric peak of 103.49 mL of $CO_2$ $kg^{-1} \cdot h^{-1}$ for the "Agüitada" genotype, collected at Estacion Nanchi, Santiago Ixcuintla, Nayarit, Mexico. A possible explanation for the values of RR of the genotypes used in this study may be that during the respiration process, the fruit has various substrates causing losses of reserves. Substrates such as starches, sugars, and organic acids can be transformed into simpler compounds such as $CO_2$ and $H_2O$. An example of this is the transformation of hexoses into $CO_2$; for every 180 g of hexose, 264 g of $CO_2$ is generated [23]. Thus, by finding differences in respiration parameters, it is possible to associate these differences with the content of some physicochemical compounds (TA and TSS), and the variability of these parameters between genotypes can be reflected in the RR [7].

In the case of differences between days of storage, the respiration rate of climacteric fruits increases at the beginning of growth and decreases at the end of the shelf life, with such behavior coinciding with the stage of ripening [38].

A characteristic of climacteric fruits is that after the appearance of the climacteric peak, maximum EP occurs [17]. Different values in EP have been found in different jackfruit genotypes. Mata-Montes de Oca et al. [15] reported an EP of 21.4 μL $kg^{-1} \cdot h^{-1}$ for jackfruit (genotype not specified) on 1 day after the appearance of the climacteric peak. A different value was reported by Morelos-Flores et al. [16] for the "Agüitada" jackfruit genotype;

the maximum EP coincided with the climacteric peak of the fruit (day 3), generating EP values of 45.55 µL kg$^{-1}$·h$^{-1}$. The fruits produce ethylene through a biosynthetic pathway in which methionine is converted to 1-aminocyclopropane-1-carboxylic acid (ACC) and then to ethylene [37]. In this pathway, the enzyme ACC oxidase (1-aminocyclopropane-1-carboxylic oxidase) is responsible for the transformation of ACC to ethylene in the last stage of this cycle. The activity of ACC oxidase is inhibited by $CO_2$ ions, µ-aminoisobutyric acid, and temperatures above 35 °C, in addition to being highly dependent on oxygen [39]. This suggests a relationship between RR and the increases in the rates of EP, because in this investigation, significant differences were found in the RR of the studied genotypes and day of storage. Another option that can be taken into account is that the fruits have established sensitivity to ethylene, because this hormone is biologically active at very low concentrations, ranging from 1 ppm to 1 ppb, triggering autocatalytic production [40].

With regard to PWL, the loss of water in the form of vapor is known as transpiration. This phenomenon occurs through the cuticle, stomata, or lenticels of the exposed area of the fruits. The loss of cell turgor is due to the fact that water in the cytoplasm moves through the membranes and intracellular spaces to the surface of the fruit [41]. This concept may suggest that the structure of the jackfruit peel in different genotypes may be formed by the same structures between materials, which is reflected by the fact that no significant differences were observed for this variable. On the other hand, as mentioned above, respiration uses starches, sugars, and organic acids as substrates that are involved in the loss of solids in PWL. Moreover, fruit structure and transpiration mainly influence this parameter [42–44], which correlates with differences between different days of jackfruit storage.

*4.3. Physicochemical Analysis*

The data obtained from PC in this study can be compared with those obtained by Morelos-Flores et al. [16] for the "Agüitada" genotype. Their investigation reported initial values of 106.25 °Hue, which can be interpreted as bright green, while at the end of shelf life, 84.45 °Hue was reported, turning the peel to olive green. The main causes of PC changes are chlorophyll degradation, which occurs after the development of oxidative processes, changes in pH, and the action of chlorophyllases (EC 3.1.1.14) during fruit ripening [22]. Moreover, it has been reported that the degradation of chlorophylls by enzymatic activity is induced in response to the production of ethylene. In addition, this hormone affects the functionality of chloroplasts due to the decrease in the fluorescence of chlorophyll [45].

In the case of BC, our findings can be compared with data reported by Balamaze et al. [46], who studied three different jackfruit genotypes from Malangala, Mityana locality, Uganda. Three different values of °Hue were reported in the pulp of the genotypes: 101.40 °Hue (yellow), 105.36 °Hue (white) and 94.92 °Hue (orange). With regard to the color of jackfruit bulbs, abscisic acid (ABA) is a precursor derivative of carotenoid C40, generating phytoene along the way, which is converted to lycopene by the action of several desaturases. Afterwards, the formation of cyclic rings and hydroxylation generates zeaxanthin to later enzymatically (zeaxanthin epoxidase) produce violaxanthin. This molecule is converted into neoxanthin by isomerases, a substrate for the enzyme 9-*cis*-epoxycarotenoid (NCED), generating xantoxin, which is finally converted into ABA [47]. At this point, the link between the generation of carotenoids and the production of ABA is evident, because the genotypes evaluated in this research that showed significant differences were those with orange coloration; β-carotene, a predecessor molecule of zeaxanthin, was predominant in those fruits.

Morelos-Flores et al. [16] reported a PF of 238.68 N for "Agüitada" in its last day on the shelf, similar to the results of our investigation. In the case of BF, Balamaze et al. [46] reported a range of firmness for three genotypes with values of 6.6 to 12.5 N at a state of maturity for consumption, different from our results. In climacteric fruits, the beginning of the ripening and softening of the components is marked by an increase in respiration and EP, the hormone that controls the genes that regulate ripening. As a result, the cell wall and the constituent polymers are progressively modified during fruit ripening, giving way to

hydration in the structure as the union between pectins, the main softening factor, changes the texture of the fruit [48,49].

Lower percentages of TA were reported by Ibrahim et al. [24]; 10 jackfruit genotypes ranged in TA from 0.045% to 0.058%, with pH values ranging from 4.55 to 4.75.

The most abundant organic acids in jackfruit are malic, citric, succinic, and oxalic acids, which may decrease in abundance during fruit ripening; therefore, changes in pH and TA content among genotypes may not be representative, but rather, they may be noticeable in the ripening process and storage days. These acids may function as a substrate for respiration and as a carbon skeleton for new compounds generated during ripening [7].

TSS values similar to those obtained in this investigation were reported by Morelos-Flores et al. [16], with 29.55 TSS for the "Agüitada" genotype on its last day of storage. Similarly, Ibrahim et al. [24] reported a range of 19.6 to 25.3 TSS, similar to the results in this study. An increase in TSS was observed in all of the studied genotypes and their storage days, which was attributed to the increase in soluble galacturonic acids as a result of the degradation of insoluble pectins found in the cell wall, caused by the enzymatic activity of polygalacturonase [7].

## 5. Conclusions

In this study, a morphological analysis of four genotypes of jackfruit showed significant differences ($p \leq 0.05$) in 17 out of 19 parameters. The "Rumina" and "Licenciada" genotypes stood out in bulb size. In the physiological analysis, differences in RR and EP among genotypes were found; "Agüitada" produced the highest rate of $CO_2$ and "Virtud" had the highest EP rate. Based on our physicochemical analyses, BC, PF, TA, and TSS showed significant differences between the genotypes and days, while PC, BF, and pH were similar in all genotypes. Based on the quality of the genotypes obtained from the previously mentioned analyses, "Rumina" and "Licenciada" stood out as fruits suitable for exportation. On the other hand, the PCA of the different days of storage helped to elucidate the behavior of the genotypes, denoting the differences among them, despite the fact that there were no differences in some physicochemical and physiological parameters. In general, this study was able to show differences among the jackfruit genotypes used in Nayarit, which could help identify alternative uses, e.g., export, commercialization of pre-cut fruit, fruit processing and postharvest management.

**Author Contributions:** Conceptualization, E.M.-G., M.A.C.-L., A.S.-V, V.M.Z.-G. and G.T.-G.; Formal analysis, D.A.M.-F., V.M.Z.-G. and M.d.L.G.-M.; Investigation, D.A.M.-F.; Methodology, D.A.M.-F.; Project administration, G.T.-G. and M.d.L.G.-M.; Supervision, E.M.-G. and M.d.L.G.-M.; Writing—review & editing, M.d.L.G.-M. All authors have read and agreed to the published version of the manuscript.

**Funding:** This research received no external funding.

**Acknowledgments:** The authors thank CONACYT (Mexico) for the scholarship to David Antonio Morelos-Flores (9348) and the Tecnológico Nacional de México/Instituto Tecnológico de Tepic for support for the development of this project.

**Conflicts of Interest:** The authors declare no conflict of interest.

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
