# Peer review of "Comparative Study of Four Jackfruit Genotypes: Morphology, Physiology and Physicochemical Characterization"

_horticulturae, doi:10.3390/horticulturae8111010_

Round 1
Reviewer 1 Report (Previous Reviewer 4)
This is the third time that I reviewed this manuscript. Although the authors have made efforts to improve the paper, it still requires additional revision to make it more acceptable and defensible. Many comments have not been addressed in this version.
Some specific comments:
L17: Replace “ Morphological analysis, physiological analysis including respiration rate, ethylene production and physiological weight loss were performed. Physicochemical analyses included color, firmness, titratable acidity, pH and total soluble solids in the peel and bulb.” By “Four genotypes, locally known namely: Agüitada, Licenciada, Rumina, and Virtud, were evaluated.”
Figure 1: PLEASE reformat the figure and put a clear title and description.
For all the tables: Keep only the means and letters
Author Response
Dear reviewer 1:
Point 1: Some specific comments:
L17: Replace “ Morphological analysis, physiological analysis including respiration rate, ethylene production and physiological weight loss were performed. Physicochemical analyses included color, firmness, titratable acidity, pH and total soluble solids in the peel and bulb.” By “Four genotypes, locally known namely: Agüitada, Licenciada, Rumina, and Virtud, were evaluated.”
this change was made in lines 17-19.
Figure 1: PLEASE reformat the figure and put a clear title and description.
this change was made
For all the tables: Keep only the means and letters
this change was made
"Please see the attachment".

Reviewer 2 Report (Previous Reviewer 5)
The statistical analysis of Table 2 should be performed again. The capital letters do not coincide with the observed differences e.g. it is not possible A and F letters being together in the same mean. The authors have to explain why data from days 10 and 11 are missing form two genotypes. The same applies to Table 3.
In PCA analysis and Figure 2, why days 3 and 7 are not depicted in the graph?
LInes 401-403: this is not true for all the tested genotypes. What about those genotypes where fluctuations were recorded (e.g. Rumina, Licenciada)?
Lines 429-433: this can be substantiated by providing fruit dry weight at the tested storage days.
Based on the results, a conclusionary sentence where the bast performing genotypes are identified, depending on fruit quality after storage, should be added in the Conclusions section.
Author Response
Response to Reviewer 2 Comments
Dear reviewer 2:
Point 1: Some specific comments:
The statistical analysis of Table 2 should be performed again. The capital letters do not coincide with the observed differences e.g. it is not possible A and F letters being together in the same mean.
Statistical analysis was reviewed and tables were corrected.
The authors have to explain why data from days 10 and 11 are missing form two genotypes. The same applies to Table 3.
On lines 199-202 and 248-251 it is explained why data from days 10 and 11 of two genotypes are missing.
In PCA analysis and Figure 2, why days 3 and 7 are not depicted in the graph?
As there are several correlated variables, it was decided to use principal component analysis (PCA) in order to reduce the dimensionality of the data. And one of the requirements to increase the explanation of the principal component is to introduce data with high variability, in this way the PCA will allow to find a pattern in the data and correlate them with each other, allowing to give a greater explanation of the behaviour of the values; because there was not much variability between days 1-3, 3-5, 5-7, 7-9 and 9-11 it was decided not to analyse the intermediate days (3 and 7).
Lines 401-403: this is not true for all the tested genotypes. What about those genotypes where fluctuations were recorded (e.g. Rumina, Licenciada)?
The behaviour described in relation to RR and physicochemical parameters (TA and TSS) is representative in climacteric fruits. However, as observed in the results, the same behaviour may not occur between genotypes. This can also be attributed to other organic compounds that act as substrates for respiration (fatty acids, amino acids, ketone bodies). This can also be related to the regulation of genes expressing enzymes that regulate metabolic pathways related to respiration. For example, phosphoenolpyruvate, ATP and NADH inhibit phosphofructokinase (PFK). In another section of glycolysis, phosphoenolpyruvate to pyruvate is regulated by pyruvate kinase, being activated by adenosine diphosphate and inhibited by adenosine triphosphate. Finally, this is evidence of physiological differences in the evaluated genotypes of jackfruit.
Lines 429-433: this can be substantiated by providing fruit dry weight at the tested storage days.
The dry weight was quantified but only of the bulbs or pulp of the jackfruit and no statistically significant differences were observed in this parameter during storage, which can be seen in the following table.
Dry weight only on bulbs
Storage days |
“Agüitada” |
“Rumina” |
“Licenciada” |
“Virtud” |
|
Dry weight (%) |
|||
1 |
67.73±0.17a |
66.45±0.93a |
67.81±1.31a |
65.99±3.56a |
3 |
68.18±0.06a |
65.47±0.14a |
70.56±5.45a |
67.76±1.25a |
5 |
66.93±0.96a |
62.02±1.8a |
68.61±2.53a |
68.15±1.43a |
7 |
66.69±1.76a |
65.68±0.63a |
74.19±2.06b |
66.84±1.92a |
9 |
68.85±0.16a |
67.14±0.28a |
72.33±1.07a |
71.69±2.48a |
11 |
- |
69.86±0.45a |
70.56±0.46a |
- |
Lowercase letters represent the effect between jackfruit genotypes, different letters indicate significant differences (P ≤ 0.05) between genotypes.
Based on the results, a conclusion sentence where the bast performing genotypes are identified, depending on fruit quality after storage, should be added in the Conclusions section.
A few lines (494-496) were added in the conclusions to address this remark
"Please see the attachment".

Round 2
Reviewer 1 Report (Previous Reviewer 4)
The manuscript was considerably improved. Please modify/edit Figure 1 according to the following recommendations.
For figure 1:
1. Change the title to "Morphological aspect of the different jackfruit genotypes in cross-section after five days of storage. (A: Agüitada genotype, B: Rumina genotype, C: Licenciada genotype, D: Virtud genotype; Storage condition: 25 ºC, 90 % relative humidity)."
2. Remove all the leaves of the jackfruit genotypes (not necessary here).
3. Edit the figure as the following schema:
A
|
B |
C
|
D |
Author Response
Dear reviewer 1:
Point 1: For figure 1:
- Change the title to "Morphological aspect of the different jackfruit genotypes in cross section after five days of storage. (A: Agüitada genotype, B: Rumina genotype, C: Licenciada genotype, D: Virtud genotype; Storage condition: 25 ºC, 90 % relative humidity)."
- Remove all the leaves of the jackfruit genotypes (not necessary here).
- Edit the figure as the following schema:
A |
B |
C |
D |
Based on the comments, the title of Figure 1 (L:159-161) was modified, the images corresponding to the leaves were removed and the figure was edited according to the suggested scheme (Figure 1).

Reviewer 2 Report (Previous Reviewer 5)
The authors have adequately addressed my comments.
Author Response
Dear reviewer 2:
Point 1: The authors have adequately addressed my comments.
Thank you, respected reviewer

This manuscript is a resubmission of an earlier submission. The following is a list of the peer review reports and author responses from that submission.
Round 1
Reviewer 1 Report
See the file attached

Author Response
Title: indicate the number of genotype to be compare
Review attended
Abstract:
Abstract must cover all relevant results according to the morphology, physiology and physicochemical aspects. It also contain a lot of repetition.
18-20 Métodologia.
All keywords are already in the title. Authors could use respiration rate, ethylene production as a keyword.
Review attended
Introduction
Line 38-42: this sentence is very long.
Review attended
Line 46-47: replace the sentence by “the seed are light brown in color with a spheroid shape and ...
Review attended
Line 57: change investigation to synonym word in order to avoid repetition.
Review attended
Line 58: what kind of use are you talking about?
Review attended
Methods
2.1. Biological Material
What part of material is placed in the container? (line 88)
Review attended in L:91
Line 92: what authors mean by “the fruit of RR and EP”?
The abbreviations RR and EP were defined in line 52
Line 90 and 99: Authors mention that physiological analysis are performed every 24h whereas the fruit were placed in the container for 30 min. this is confusing. Authors should clarify it.
Review attended
the idea was clarified on the lines 93-95 y 108 -110
Line 106: replace “all in jackfruit” by using jackfruit.
Review attended L:117
Briefly explain the Firmness method
Review attended
Extended on L:120
Line 110: ... determine the color “of” the shell and bulbs.
Review attended L:122
Line 115: replace “...with a Tukey test...” by ...following with ...
Review attended L:126
Line 116: what initial, intermediate and final storage referred for?
Review attended L:127
the initial (day 1),
intermediate (day 5)
final storage day (days 9 and 11)
Results
General points: authors must present the results in several paragraph for each section according the vegetal material part such as leaf, bulb etc. they also should use a transition word to make the manuscript understandable.
recommendation followed throughout the text
3.1. Morphological Analysis
Line 128: “Agüitada” and “Licenciada” differed from the rest of the genotypes. What are this sentence used for?
It refers to the fact that the genotypes mentioned are different from the rest in the evaluated parameter; and it was written in lines 139-144.
Line 129-130: The LPE had a maximum value of 33.24 cm for “Rumina” and was statistically equal 129 to “Virtud”. “Licenciada” had a minimum value of 19.22 cm for LPE. “must be improved as Regardong LPE, Rumina and Virtud gave the highest value (..cm) while the lowest value was found with Licenciada (.. cm).
Review attended
Restructured from lines 145 to 171
Line 132: Put a transition word between genotype and Aguitada such while or whereas etc.
Review attended
Restructured from lines 139 to 144
Line 133: Put transition word in the beginning of the next sentence.
Review attended
Restructured from lines 145 to 171
Line 136: remove “the rest ... similar” because “only” already mention it.
Restructured from lines 139 to 171
Line 138: “Agüitada” and “Licenciada” were similar (in RW) and “Agüitada” and “Virtud” 138 were also similar (in RW) must be improved. Restructured from lines 139 to 171
Line 139-142: however “Rumina” and “Licenciada” showed the highest values for BW (5.4 and 4.58 kg, and BWI (......) respectively, while “Agüitada” and “Virtud” were similar for both genotype. Restructured from lines 139 to 171
Line 142: As far as concerning NSPF, not significant difference was found, however, Rumina” showed the highest value (130.33 seeds).
Restructured from lines 139 to 171
Line 148-149: Authors should firstly indicate if the difference is significance before showing la lowest or highest. If the difference is not significance, it is not necessary to explain this difference.
Restructured from lines 139 to 171
Line 150-151: Authors need to give the reference
Restructured from lines 139 to 171, references L:144
Line 151-154: idem as line 148-149.
Restructured from lines 139 to 171
However, it is considered that the wording in this part is correct since it is mentioned that the only genotype that has significant differences is the "Agüitada" genotype.
3.2. Physiological Analysis
Line 159-160: The RR” was 53.16, 41.53 and 31.74 mL of CO2 kg−1·h−1 for “Agüitada, Virtud”and “Licenciada” respectively. Review attended L:178
Line 161-168: Authors must compare the day of the climacteric peak of the genotype before moving to RR and EP. Review attended L:174-178
Line 172-177: idem as line 148-149
Restructured from lines 195 to 197
Line 180: what is “I”?
Review attended L:199
Line 182: what was the duration of storage of authors 12 and 13?
In the article by Mata-Montes de Oca et al. 2007 [12] it was 10 days of shelf life at 20 ± 1 ºC and 85 ± 5 % RH and for Morelos-Flores et al. 2021 the shelf life of jackfruit was 8 days at 25 ± 1 ºC and 85 ± 5 % RH.
Included in L:200-202
Line 183-184: what authors mean by “an acceptable percentage was obtained for these fruits”? When comparing the results obtained in this research with those obtained by authors 12 and 13, we observed a decrease in FWL; therefore, we report this parameter within the range and consider it acceptable for jackfruit.
3.3. Physicochemical Analysis
Line 186-187: from Table 2, physicochemical analyses including bulb color (BC), peel firmness (PF), and TA showed significant differences (P ≤ 0.05).
Review attended
Line 188-193: Authors jump from BC, PF and TA to PC which is not mentioned before. In addition, authors must organize the paragraph from the first day to the last day.
Restructured from lines 205-211
Line 194-196: On what basis authors divized the genotype in 2 groups? They have to explain this in this section.
Based on the color of its pulp (orange and yellow); L:215
Line 194-203: How could authors relate hue value to the color? Authors could indicate the percentage of loss of hue value and explain why?
The data obtained from BC were reported in the LCH system, in which L is set as lightness, C represents saturation and Hue is the hue angle. The °Hue allows us to find the variation of hue in a specific color, thus being able to correlate the increase of this value with the generation of phytochemical compounds such as alpha and beta carotene in jackfruit, these compounds produce yellow and orange coloration respectively; research already conducted with jackfruit reports only °Hue values to express these results (Mata-Montes de Oca et al., 2007 and Morelos-Flores et al., 2021).
Part of this, it has been included from line 223-228
Line 205-206: what authors mean by “...in only one of the genotypes studied?
Review attended
The genotype to which it refers was indicated, L:229
Line 206: Between usually indicates a range and not a sequence of numbers
Review attended, L:229-230
Line 207: close the bracket for ... and 409.15 N (“Rumina”);
Review attended L:231
Line 209-212: Authors did not mention the significant difference in BF at initial storage state. Thus how could authors explain similar behavior at the end of the storage?
Review attended and text restructured L:229-237
Line 213: It is significant difference instead of statistical difference.
Review attended L:238
Line 213-219: the genotype did not generate data. Authors must improve this paragraph using simple sentences and structuring their ideas.
Review attended and supplemented text L:238-244
Line 220-226: Authors mention that significant differences (P ≤ 0.05) were found in one of the jackfruit genotypes evaluated whereas all values cited below are different. What authors mean by “in one of jackfruit genotype”?
Review attended and text restructured L:246-250
3.4. Principal Component Analysis
This section should be rewritten. The paragraph is not comprehensive. The authors should first specify the different projections they make and then relate the variables to the projection because on the figures some scripts are unreadable. What authors mean by “the fruit L/W ratio, seed L/W ratio, PC, BC, and PF did not present important changes” and the pH, BF, leaf L/W ratio, and RR decreased in this analysis? How can PCA indicate a decreasing or increasing state of a variable?
This review was addressed and the principal component analysis section was rewritten
The variable projections on the PCA depend on the representativeness of the variable. When the variable is on the circle, it means that its projection is good and not high or low conversely.
Review attended and text restructured L:258-305
Why do the authors state in the results section that larger fruits will have a lower RR and L/W ratio? Authors should present the figure so that we can understand the data connection between them. Review attended and text restructured L:258-305
In this section, Authors indicate Y axis which is not shown in any Figure 3. Authors should relate Figure A, C, D with their opposite B, D and E for better comprehension. Review attended and text restructured L:258-305
How can PCA 2 affect the variable values? Review attended
Table 1: No WP find in tis table. This was a drafting error, it was not the abbreviation WP but WS. Corrected
- Discussion
Authors should define each abbreviation at the beginning of each section. In addition, authors are not working on the change of parameters but on the comparison of these parameters between 4 genotypes. Therefore, they have to link their explanation to this objective.
With regard to the commentary on abbreviations, these are defined in the body of the manuscript the first time they are used and then only referred to.
The discussion focuses mainly on the behaviour with respect to the different genotypes as this is the effect, which in the objective is of interest; with respect to the variability of the parameter itself with respect to time although it is seen, it is to a certain extent predictable that the days of storage would affect it, therefore there is more interest in the analysis between the genotypes.
4.1. Morphological Analysis
Authors should structure the paragraph according to each variable
Review attended
Line 317: what authors mean by” Different figures were reported by Ibrahim et al. [17]” and WP? Review attended and supplemented text L:318-322
Line 321: What is TP ? Tickness of the peel, is indicated in line 83
Line 339-344: this paragraph should be written as “Of the above-mentioned, there is statistical evidence that variability between the jack-fruit genotypes is evident in the parameters evaluated in this research (Anu et al. [23], Mitra & Maity [24], Rahman et al. [25], Rai et al. [26], Reddy et al. [27], and Singh 343 et al. [28]). This variability in the morphological attributes of fruits, especially in jackfruit, could be due to cross-pollination and different agroclimatic conditions, resulting in new variability [9]”. Review attended
L340-345
4.2. Physiological Analysis
Line 350: Who or what...”in the same state and the same country” are they referring to? Review attended and text restructured L:356-359
Line 346-356: Authors should structure the paragraph as: Respiration definition and goal, then compare their results to the literature and finish by the explanation of the variability. Besides, Authors should provide an explanation for the significant difference found in his study concerning EP based on the literature. Review attended and text restructured L:347-352 y L:356-359.
The EP data were correlated with those obtained in RR as mentioned in the literature. This is mentioned in lines 380-382.
374-380: Fruits produce ethylene through a biosynthetic pathway in which methionine is converted to 1-aminocyclopropane-1-carboxylic acid (ACC) and then to ethylene. In this pathway, the enzyme ACC oxidase (1-aminocyclopropane-1-carboxylic acid oxidase) is responsible for the transformation of ACC to ethylene in the last stage of this cycle. ACC oxidase activity is inhibited by CO2 ions, µ-aminoisobutyric acid and temperatures above 35 °C, and is highly oxygen-dependent.
Moreover, RR and EP can’t have a behavior. This is not proper English. The word was changed to values
Line 368-384: Authors should provide an explanation for the significant difference found in his study concerning EP based on the literature. The EP data were correlated with those obtained in RR as mentioned in the literature. This is mentioned in lines 380-382.
374-380: Fruits produce ethylene through a biosynthetic pathway in which methionine is converted to 1-aminocyclopropane-1-carboxylic acid (ACC) and then to ethylene. In this pathway, the enzyme ACC oxidase (1-aminocyclopropane-1-carboxylic acid oxidase) is responsible for the transformation of ACC to ethylene in the last stage of this cycle. ACC oxidase activity is inhibited by CO2 ions, µ-aminoisobutyric acid and temperatures above 35 °C, and is highly oxygen-dependent.
Line 385-391: idem As no significant statistical differences were obtained, these results are attributed to the structure of the jackfruit, which can be theorised to be the same.
4.3. Physicochemical Analysis
Line 393-396: the sentence is very long and therefore incomprehensible. For what respectively is used for? Review attended and text restructured L:392-400
Line 393-401: why is there a significant difference between the genotype studied? Also, this paragraph must merge into one. An editing error was detected in the table and the Hue values at the end of shelf life indicate that there are no significant differences between the genotypes. Therefore, the discussion focuses on why the change in fruit skin colouring occurs.
Line 402-405: Authors must indicate the HUE value which will be related to the color, not the opposite. Review attended L:403-408
Line 405-409: the sentence is long. Review attended
Line 402-414: Same observation for BC. See Comment for the Line 393-401. Review attended and text restructured L:392-400
Line 416-437: Idem, See Comment for the Line 393-401. Review attended and text restructured L:392-400. In the sections indicated, the reasons for the physico-chemical changes of these variables are discussed, where it is implied that they will be dependent on the ethylene production, respiration and enzyme activity of each genotype.
Conclusion
Is Authors measure the content of bulbs? The number of bulbs was counted, however they were not reported in this manuscript. This was a typing error
What authors mean by “ the most CO2”? Review attended
Line 444-447: this sentence is confusing. Review attended and supplemented text
Line 449: Could authors indicate an example of the utilization they are talking about? Review attended and supplemented text

Reviewer 2 Report
This manuscript focused on jackfruit genotypes. But the paper just like a popular science article. The quality of figures are low, just like Fig. 1, there were no ruler in the figures, and the labels under the fruit could not be seen clearly. And the respiration rate were too high, I think this data may be have some problems.
Author Response
Este manuscrito se centró en los genotipos de jackfruit. Pero el documento es como un artículo de divulgación científica.
La calidad de las figuras es baja, al igual que la Fig. 1, no había regla en las figuras y las etiquetas debajo de la fruta no se podían ver claramente. The photos of fruits and leaves could be relabelled, but the authors would like to propose to the editor to consider the digital photos as accompanying material so that they can be clearly seen when enlarged.
Y la tasa de respiración era demasiado alta, creo que estos datos pueden tener algunos problemas. Jackfruit is a fruit with a high respiratory rate and high ethylene production concentration, which is supported in other publications (Mata-Montes de Oca et al., 2007; Morelos-Flores et al., 2021) related to RR and values close to those found in our research are reported.

Reviewer 3 Report
Review Comparative study of jackfruit genotypes: morphology, physiology and physicochemical characterization
The paper discusses the properties of 4 Jackfrut genotypes grown in Mexico. The quality parameters and storage potential of fruits of this species are compared. The work is constructed correctly in the general outline of the method and the method of conducting the experiment has been selected in the correct way. Nevertheless, some minor errors, both linguistic and substantive, have not been avoided. Details are given below.
The abstract exceeds the permissible number of 200 words , but if the academic editor accepts it not exceeding 10% increase in text, I will not have any comments either.
Line 23: skip ”either”
Line 25: broad → a wide
Line 37: what you mean traditional? Mango is as well native to Asia
Line 45: making up → comprising
Line 48: Information always without the
Line 51: skip “that are”
Line 62-66: Please describe the procedure that was used to ensure that the fruits collected for testing are representative of the genotype.
Line 66: How physiological maturity was determined.
Line 84: Vernier is a name!
Line 90: in → on
Line 120: Analyses of variance were carried using → Analyses of variance was carried out using
Results: General note: Repeating the results in the tables in the text is unnecessary.
Figure3: Please expand all abbreviations that have been used in PCA charts
Lines 313-344: In the Discussion on morphological analysis of genotypes, there is no summary that would tell what significance these features have from the point of view of the cultivation of this species. Especially those whose value differs from previous measurements obtained by other researchers.
Line 346: Data on the RR of jackfruit is → Data on jackfruit RR are available from
In the conclusions, I lack some general summary that would show how significant these results are and which ecotypes from the subjects will be important growing in the cultivation of jackfruit in Mexico. The recommendations that the authors could give for production would also be valuable.
Author Response
The paper discusses the properties of 4 Jackfruit genotypes grown in Mexico. The quality parameters and storage potential of fruits of this species are compared. The work is constructed correctly in the general outline of the method and the method of conducting the experiment has been selected in the correct way. Nevertheless, some minor errors, both linguistic and substantive, have not been avoided. Details are given below.
The abstract exceeds the permissible number of 200 words, but if the academic editor accepts it not exceeding 10% increase in text, I will not have any comments either. In the authors' guide they mark 200 words, we comply with this requirement.
Line 23: skip ”either” Review attended
Line 25: broad → a wide Review attended
Line 37: what you mean traditional? Mango is as well native to Asia The word "traditional" refers to a crop that has been managed for a long time and is therefore well established in terms of production and utilization. Considering this, jackfruit is a relatively young and exotic crop in our country.
Line 45: making up → comprising Review attended
Line 48: Information always without the Review attended
Line 51: skip “that are” Review attended
Line 62-66: Please describe the procedure that was used to ensure that the fruits collected for testing are representative of the genotype. In the preliminary tests, two orchards were visited in the state of Nayarit, one located in "Las Varas", a town in the municipality of Compostela and the other in "Estación Nanchi", a town in the municipality of Santiago Ixcuintla.
The test was carried out on 5 trees per accession, giving a total of 15 fruits and 50 leaves per accession.
The length and width of the leaves and the polar and equatorial diameter of the fruits were measured. Subsequently, the results were processed and handled in the statistical program version
12 for the calculation of the sample size by means of power analysis.
Line 66: How physiological maturity was determined. This part was carried out at the moment of cutting the fruit; as a first point, it was considered that a "hollow" sound was generated by gently tapping the fruit. And as a second point, when causing a wound to the stalk, the time it takes for latex to stop sprouting from the wound was measured (time less than 3 min).
Line 84: Vernier is a name! Review attended
Line 90: in → on Review attended
Line 120: Analyses of variance were carried using → Analyses of variance was carried out using Review attended
Results: General note: Repeating the results in the tables in the text is unnecessary. the data that could be repeated were those necessary for comparison and subsequent discussion.
Figure3: Please expand all abbreviations that have been used in PCA charts Review attended
Lines 313-344: In the Discussion on morphological analysis of genotypes, there is no summary that would tell what significance these features have from the point of view of the cultivation of this species. Especially those whose value differs from previous measurements obtained by other researchers.
This research is making an incursion in obtaining morphological data that will characterise the genotypes of yaca in the country, due to the fact that there is little or no information on these parameters, which is of utmost importance because it will give elements to start with the processes of registration of varieties for the protection of these fruits
Line 346: Data on the RR of jackfruit is →Data on jackfruit RR are available from Review attended
In the conclusions, I lack some general summary that would show how significant these results are and which ecotypes from the subjects will be important growing in the cultivation of jackfruit in Mexico. The recommendations that the authors could give for production would also be valuable. The analyses carried out revealed that the Rumina and Licenciada genotypes are the ones that can be handled mainly as export fruits. Furthermore, the characterisation of the four genotypes studied may allow the registration and protection of these fruits that have been developed in the region.

Reviewer 4 Report
INTRODUCTION:
The introduction section needs to be more developed; the authors should add some information about the effect of climate, soil and cropping systems on Jackfruit physicochemical morphological features. It is also of great interest to report information about the functional, medicinal, and therapeutic qualities and benefits of jackfruit. And link such information with the genotypes/varieties….
L33 “Due to its edaphoclimatic adaptability, ….. characteristics of the fruit.” L38, should be moved to the last part of the introduction L54.
L40: “ . Moreover,….”
MATERIAL AND METHODS
L61: the section “Biological Material” should be rewritten as:
Remove the citation of the figure (should be shown in the result).
Put firstly the genotype's names, the region where those genotypes are grown, the season of sampling (February 2022), and if possible some features of the climate and soil… Then “ For the morphological analyses….. For the physiological and physicochemical……
L87: Add RR and EP in full letter
RESULT:
Authors should use correctly the MDPI manuscript template. Tables and figures must be included in the main text !!
Figure 1: Please reformed the figure and put a clear title and description
Tables: Keep only means and letters
Morphological Analysis
Rapport here results of leaves morphology (Figure 1), I suggest removing this analysis from all the paper
L132: Remove “Currently, there are no reports regarding LPE and DEPE of jackfruit in national and international databases”!
L139: Replace “genotypes. However, …”
L145: Remove “Like LPE and DEPE, WSB is ….”!
ABSTRACT:
L14: replace; by point “. However,….”
L17 replace; by point “. Four ….”
“Four genotypes, locally known namely: Agüitada, Licenciada, Rumina, and Virtud, were evaluated.
Author Response
The introduction section needs to be more developed; the authors should add some information about the effect of climate, soil and cropping systems on Jackfruit physicochemical morphological features. It is also of great interest to report information about the functional, medicinal, and therapeutic qualities and benefits of jackfruit. And link such information with the genotypes/varieties….The effect of climate was included in the introduction. The functional part has been worked on, however this information will be considered for another manuscript.
L33 “Due to its edaphoclimatic adaptability, ….. characteristics of the fruit.”
L38, should be moved to the last part of the introduction L54. it was considered to maintain the order of the paragraph
L40: “ . Moreover,….” Review attended
MATERIAL AND METHODS
L61: the section “Biological Material” should be rewritten as: In various manuscripts, the section "biological material" is used, as in this case, to describe the genotypes used.
Remove the citation of the figure (should be shown in the result). The figure is not a result, it was considered in materials and methods as the leaves and fruit of the different genotypes are study materials and are illustrative as images.
Put firstly the genotype's names, the region where those genotypes are grown, the season of sampling (February 2022), and if possible some features of the climate and soil… Then “ For the morphological analyses….. For the physiological and physicochemical……these specifications can be seen in lines 65-73
L87: Add RR and EP in full letter Review attended
RESULT:
Authors should use correctly the MDPI manuscript template. Tables and figures must be included in the main text !! Review attended
Figure 1: Please reformed the figure and put a clear title and description Review attended
Tables: Keep only means and letters Review attended
Morphological Analysis
Rapport here results of leaves morphology (Figure 1), I suggest removing this analysis from all the paper This parameter is used in the research of Biswajit&Kartik, 2019; Chhetri et al, 2019; Phaomei et al, 2017, so we consider it important in the morphological characterisation of jackfruit.
L132: Remove “Currently, there are no reports regarding LPE and DEPE of jackfruit in national and international databases”! Review attended
L139: Replace “genotypes. However, …” Review attended
L145: Remove “Like LPE and DEPE, WSB is ….”! Review attended
ABSTRACT:
L14: replace; by point “. However,….” Review attended
L17 replace; by point “. Four ….” “Four genotypes, locally known namely: Agüitada, Licenciada, Rumina, and Virtud, were evaluated. Review attended

Reviewer 5 Report
In the present manuscript, a morphological, physiological, and physicochemical characterization of four jackfruit genotypes cultivated in Mexico was performed.
Introduction is too short. More information is needed regarding the species and the studied parameters.
The statistical analysis in Table 2 should be performed with a two-way ANOVA (storage time x genotype).
Figure 2 should be replaced by a Table with the proper statistical analysis. On what basis is expressed the weight loss (fresh or dry weight)?
Figure 3: a multivariate analysis with two factors should be performed.
Author Response
In the present manuscript, a morphological, physiological, and physicochemical characterization of four jackfruit genotypes cultivated in Mexico was performed.
Introduction is too short. More information is needed regarding the species and the studied parameters. Review attended
The statistical analysis in Table 2 should be performed with a two-way ANOVA (storage time x genotype). A single factor analysis was used because the aim of the research is to study the differences between genotypes and not the effect of days of storage on the jackfruit.
Figure 2 should be replaced by a Table with the proper statistical analysis. On what basis is expressed the weight loss (fresh or dry weight)? The physiological behaviour of the fruit is usually reported by means of graphs as Mata-Montes de Oca et al. (2007), Morelos-Flores et al. (2021), Vargas-Torres et al. (2017), because it is easier to visualise the fruit respiration behaviour and the appearance of the climacteric peak.
Regarding the PWL, it is taken from whole fruit and the difference in weights during storage from one day to the next.
Figure 3: a multivariate analysis with two factors should be performed. The main interest of this research was to find differences between four genotypes of yaca from the state of Nayarit, not the effect of storage days. Therefore, it was only important to evaluate the morphological, physiological and physicochemical characteristics of the four genotypes in the initial, middle and final days of their storage life.

Round 2
Reviewer 2 Report
The authors had addressed all the comments and improved the manuscript. And I have no questions now.
Author Response
Response to Reviewer 2 Comments
Dear reviewer 2:
Point 1: Review report form
Response 1
Please be informed that the introduction was enriched and references were added both in the introduction and in the body of the text.
We also improved the description of some methodologies and rewrote some of the results.
In the corrected manuscript you will see the modifications highlighted in blue.

Reviewer 4 Report
The authors did not respond appropriately to reviewers' comments! The manuscript is unsuitable for publication in this version. Authors must carefully and accurately address the reviewers' comments/suggestions. The manuscript might be resubmitted after a thorough revision taking into account all the reviewers' reports.
Author Response
Response to Reviewer 4 Comments
Dear reviewer 4:
Point 1: Review report form
Response 1
Please be informed that the introduction was enriched and references were added both in the introduction and in the body of the text.
We also improved the description of some methodologies and rewrote some of the results.
Point 2: From the above comments:
The introduction section needs to be more developed; the authors should add some information about the effect of climate, soil and cropping systems on Jackfruit physicochemical morphological features. It is also of great interest to report information about the functional, medicinal, and therapeutic qualities and benefits of jackfruit. And link such information with the genotypes/varieties
L33 “Due to its edaphoclimatic adaptability, characteristics of the fruit.”
L38, should be moved to the last part of the introduction L54.
L40: “ . Moreover,….”
Response 2
In response to this observation, the information corresponding to lines L:34 to L:47 and L:61 to L:67 was included, and the wording of lines L:72-L:75 and L:76-L:80 was modified.
In the corrected manuscript you will see the modifications highlighted in blue.

Reviewer 5 Report
The authors provided their responses to my comments. However, the manuscript still needs to be revised before it is suitable for publication.
Introduction is too short. More information is needed regarding the species and the studied parameters.
Response: Review attended
New comment: the introduction is still too short. Some more details about the postharvest physiology and how they affect quality of jackfruit or similar fruit would be useful.
The statistical analysis in Table 2 should be performed with a two-way ANOVA (storage time x genotype).
Response: A single factor analysis was used because the aim of the research is to study the differences between genotypes and not the effect of days of storage on the jackfruit.
New comment: If the aim was only to characterize the genotypes why you performed a storage experiment? You should present only the results related to characterization.
Figure 2 should be replaced by a Table with the proper statistical analysis. On what basis is expressed the weight loss (fresh or dry weight)?
Response: The physiological behaviour of the fruit is usually reported by means of graphs as Mata-Montes de Oca et al. (2007), Morelos-Flores et al. (2021), Vargas-Torres et al. (2017), because it is easier to visualise the fruit respiration behaviour and the appearance of the climacteric peak. Regarding the PWL, it is taken from whole fruit and the difference in weights during storage from one day to the next.
New comment: You should try to visualize the statistical differences (using asterisks or Latin letters). Regarding the PWL, what was the basis of weight calculation? The physiological loss implies weight losses due to respiration and consumption of solids, which different from weight loss due to transpiration.
Figure 3: a multivariate analysis with two factors should be performed. The main interest of this research was to find differences between four genotypes of yaca from the state of Nayarit, not the effect of storage days. Therefore, it was only important to evaluate the morphological, physiological and physicochemical characteristics of the four genotypes in the initial, middle and final days of their storage life.
New comment: If the aim was only to characterize the genotypes why you performed a storage experiment? You should present only the results related to characterization.
Author Response
Response to Reviewer 5 Comments
Dear reviewer 5:
Point 1: Review report form
Response 1
Please be informed that the introduction was enriched and references were added both in the introduction and in the body of the text.
We also improved the description of some methodologies and rewrote some of the results.
Point 2.
Introduction is too short. More information is needed regarding the species and the studied parameters. New comment: the introduction is still too short. Some more details about the postharvest physiology and how they affect quality of jackfruit or similar fruit would be useful.
Response 2
In response to this observation, the information corresponding to lines L:34 to L:47 and L:61 to L:67 was included and the wording of lines L:72-L:75 and L:76-L:80 was modified.
Point 3.
The statistical analysis in Table 2 should be performed with a two-way ANOVA (storage time x genotype). New comment: If the aim was only to characterize the genotypes why you performed a storage experiment? You should present only the results related to characterization.
Response 3
A two-way analysis was performed L:144-151, which corrected table 2. Some lines were also modified on the basis of this analysis,
L:206-L:209,
L:229-L:232
L:236-L:240
L:268-L:269
L:273-L:277
L:282-L:287
L:290-L:296
Point 4.
Figure 2 should be replaced by a Table with the proper statistical analysis. On what basis is expressed the weight loss (fresh or dry weight)? . New comment: You should try to visualize the statistical differences (using asterisks or Latin letters). Regarding the PWL, what was the basis of weight calculation? The physiological loss implies weight losses due to respiration and consumption of solids, which different from weight loss due to transpiration.
Response 4
A table was made in place of the figure and the wording for PWL was improved, the equation used was included in the methodology section L:122-L:127; and in the discussion part of this result L:434-438 it is stated that "Physiological loss involves weight losses due to respiration and solid consumption, which are different from weight losses due to transpiration" ...............
Point 5
Figure 3: a multivariate analysis with two factors should be performed. The main interest of this research was to find differences between four genotypes of yaca from the state of Nayarit, not the effect of storage days. Therefore, it was only important to evaluate the morphological, physiological and physicochemical characteristics of the four genotypes in the initial, middle and final days of their storage life. New comment: If the aim was only to characterize the genotypes why you performed a storage experiment? You should present only the results related to characterization
Response 5
The multivariate analysis was reformulated to two factors as indicated in the methodology in the statistical analysis part L:143-151 and the result can be seen in the lines, L:306-L:347,
Part of the text L:401-L:403 was also modified.
L:429-L433
L:470-L:482
In the corrected manuscript you can see the modifications highlighted in blue.
